# A mouse model that is immunologically tolerant to reporter and modifier proteins

Kaspar Bresser [1,8], Feline E. Dijkgraaf[1,8], Colin E. J. Pritchard[2], Ivo J. Huijbers [2], Ji-Ying Song[3], Jan C. Rohr[4,5], Ferenc A. Scheeren [6] & Ton N. Schumacher [1,7 ✉]

Reporter proteins have become an indispensable tool in biomedical research. However, exogenous introduction of these reporters into mice poses a risk of rejection by the immune system. Here, we describe the generation, validation and application of a multiple reporter protein tolerant 'Tol' mouse model that constitutively expresses an assembly of shuffled reporter proteins from a single open reading frame. We demonstrate that expression of the *Tol* transgene results in the deletion of CD8[+] T cells specific for a model epitope, and substantially improves engraftment of reporter-gene transduced T cells. The Tol strain provides a valuable mouse model for cell transfer and viral-mediated gene transfer studies, and serves as a methodological example for the generation of poly-tolerant mouse strains.

[1] Division of Molecular Oncology & Immunology, Oncode Institute, The Netherlands Cancer Institute, Amsterdam, The Netherlands. [2] Mouse Clinic for Cancer and Aging research (MCCA) Transgenic Facility, The Netherlands Cancer Institute, Amsterdam, The Netherlands. [3] Animal Pathology, The Netherlands Cancer Institute, Amsterdam, The Netherlands. [4] Center for Chronic Immunodeficiency, Medical Center, Faculty of Medicine, University of Freiburg, Freiburg, Germany. [5] Center for Pediatrics and Adolescent Medicine, Medical Center, Faculty of Medicine, University of Freiburg, Freiburg, Germany. [6] Department of Medical Oncology, Leiden University Medical Center, Leiden, The Netherlands. [7] Department of Immunohematology and Blood Transfusion, Leiden University Medical Center, Leiden, The Netherlands. [8] These authors contributed equally: Kaspar Bresser and Feline E. Dijkgraaf. ✉email: t.schumacher@nki.nl

Following the advent of standardized genetic editing techniques, researchers have isolated a large collection of reporter and modifier proteins (RPs and MPs, respectively) from a variety of species that have since been instrumental to characterize a wide range of biological processes[1]. For example, RPs are frequently used to tag endogenous proteins or to track the behavior of individual cells in vivo[2,3]. In addition, MPs (e.g., Cre recombinase and Cas9) are commonly applied to influence in vivo cell behavior through the induction of defined genetic alterations[4]. However, as RPs and MPs are almost invariably derived from non-mammalian species, exogenous introduction of these proteins into immunocompetent mice poses the risk of immunological rejection by host-derived T cells. In line with this, multiple cases of immunological rejection of cells expressing RPs, such as firefly luciferase and eGFP, have been reported[5–7]. Moreover, major histocompatibility complex (MHC) class I-restricted epitopes of luciferase and GFP have been identified, underlining their capacity to induce CD8$^+$ T-cell responses[5–8]. Even in the absence of complete immunological rejection of RP- or MP-modified cells, experimental outcomes may potentially be subtly biased through the action of such undesirable immune responses.

To overcome this problem, we set out to engineer a transgene that—once introduced into the genome of a mouse model of choice—prevents the generation of immune responses against RP- or MP-modified cells through the physiological self-tolerance mechanisms. Specifically, CD4$^+$ and CD8$^+$ T cells that carry a self-reactive T-cell receptor (TCR) may be inactivated or deleted through multiple mechanisms. First, during their maturation in the thymus, T cells that carry a self-reactive TCR are deleted through negative selection. Second, auto-reactive T cells that escape deletion through central tolerance are kept in check by a mechanism referred to as peripheral tolerance that requires antigen encounter outside of the primary lymphoid organs[9]. In addition, self-reactive CD4$^+$ T cells may develop into regulatory T cells, and play a crucial role in such peripheral tolerance[10]. Thus, in order to generate a poly-tolerant mouse model, constitutive expression of foreign antigens in both the thymus and throughout peripheral tissues would be preferred. However, organism-wide expression of proteins such as eGFP or Cre is incompatible with their intended use as cell-type specific reporters or modifiers.

## Results

### Generation and validation of the 'Tol' mouse model.
In order to engineer a multiple reporter protein tolerant mouse model, we designed a large chimeric open reading frame (ORF) that encodes 6 fluorescent proteins, firefly luciferase, and Cre-recombinase, all in a scrambled format. Specifically, in order to prevent functional expression of the introduced proteins, each individual gene was split into two fragments, and the resulting set of 16 gene fragments was subsequently assembled in a scrambled order. In addition, to ensure tolerance toward potential T-cell epitopes present at the gene breakpoints, the 60 base pair region surrounding each split site was added (Fig. 1a). Finally, to be able to test whether tolerance was induced against epitopes throughout the artificial protein, a control CD8$^+$ T-cell epitope derived from the HPV E7 protein (HPV E7$_{49–57}$) was placed at the COOH-terminus. This resulted in a 7359 base pair chimeric 'Tol' ORF, encoding a protein of ±275 kDa that covers seven RP and one MP (Supplementary Fig. 1).

Recognition of RP- and MP-derived epitopes is dependent on the host MHC haplotype[11,12], and risk of immunological rejection of cells modified with individual RPs and MPs thus varies between mouse strains. To test if the Tol cassette could be applied to induce tolerance against proteins of interest, we set out to assess its functionality in the C57BL/6 strain, a widely used mouse strain in immunological research. Notably, MHC binding predictions indicated that the Tol protein contains 78 potential MHC ligands predicted to bind with high affinity (i.e., netMHC4.0 percentile rank < 1%) to the C57BL/6 MHC haplotypes (Supplementary Data 1). To establish the transgenic C57BL/6 strain, the Tol gene was targeted to the Col1a1 locus of embryonic stem cells (ESCs) via recombinase-mediated cassette exchange (Fig. 1b)[13,14]. Successfully modified ESCs were then injected into blastocysts and transferred to pseudopregnant foster mice. The resulting F0 generation was tested for transgene presence (Supplementary Fig. 2a), and mice with sufficient chimerism were bred to obtain experimental cohorts of heterozygous ('Tol') and wild-type littermate controls ('WT').

Due to its large size and scrambled design, the Tol protein is expected to be unfolded. Cellular stress induced by accumulation of unfolded proteins has previously been shown to be associated with a variety of pathologies and neurological disorders[15]. To test whether expression of the Tol transgene would affect murine development, histopathological evaluation of tissues from Tol and WT mice was performed, revealing no evidence for transgene-induced pathology (Supplementary Table 1). As a second test for cellular stress, or potential deregulation of other pathways, that may be induced by Tol expression, RNA-sequencing was performed on cerebral and cerebellar tissue of Tol and WT animals. In addition, transcriptional activity in thymic tissue, the site of T-cell deletion, was compared between Tol-transgenic and WT mice. Hierarchical clustering analysis of the top expressed genes across tissues showed that WT and Tol mice did not typically cluster separately, suggesting that the Tol transgene did not induce major transcriptional changes (Supplementary Fig. 2b). Subsequent differential gene expression analysis likewise revealed no significant differentially expressed genes, aside from two marginally downregulated genes in the thymus (Fig. 1c and Supplementary Data 2). Importantly, the Tol transcript could be detected in all organs tested (Fig. 1c, d), and aligned reads spanned the entire transgene, demonstrating that the full-length ORF was transcribed (Supplementary Fig. 2c).

### Immunological tolerance toward the E7$_{49–57}$ epitope in Tol mice.
Having observed the presence of the full-length Tol transcript in thymic tissue of Tol mice, we next investigated whether T-cell tolerance toward epitopes encoded by the transcript was successfully induced. To this purpose, WT and Tol animals were vaccinated with a DNA vector encoding the HPV E7$_{49–57}$ epitope, and the development of CD8$^+$ T-cell responses against the E7$_{49–57}$ epitope was subsequently monitored in peripheral blood by peptide-MHC multimer staining (Fig. 2a). Following DNA vaccination, a large population of E7$_{49–57}$-specific CD8$^+$ T cells was detected in blood of WT mice, with peak E7$_{49–57}$-specific CD8$^+$ T-cell frequencies up to 30% of the total CD8$^+$ T-cell population (Fig. 2b, c). In stark contrast, no E7$_{49–57}$-specific CD8$^+$ T cells were detected above background in Tol-transgene positive animals at any timepoint post vaccination (Fig. 2b, c). These data indicate strict tolerance of Tol mice toward a T-cell epitope that is present at the 3′ of the Tol ORF, indicating full-length translation of the protein and successful processing by the antigen presentation machinery.

### Enhanced engraftment of OT-I T cells that express fluorescent reporter proteins in Tol animals.
To assess whether expression of the shuffled reporter gene fragments in Tol mice resulted in tolerance toward the parental non-shuffled reporter proteins, CD8$^+$ Ly5.1$^+$ OT-I T cells—specific for the OVA$_{257–364}$ peptide—

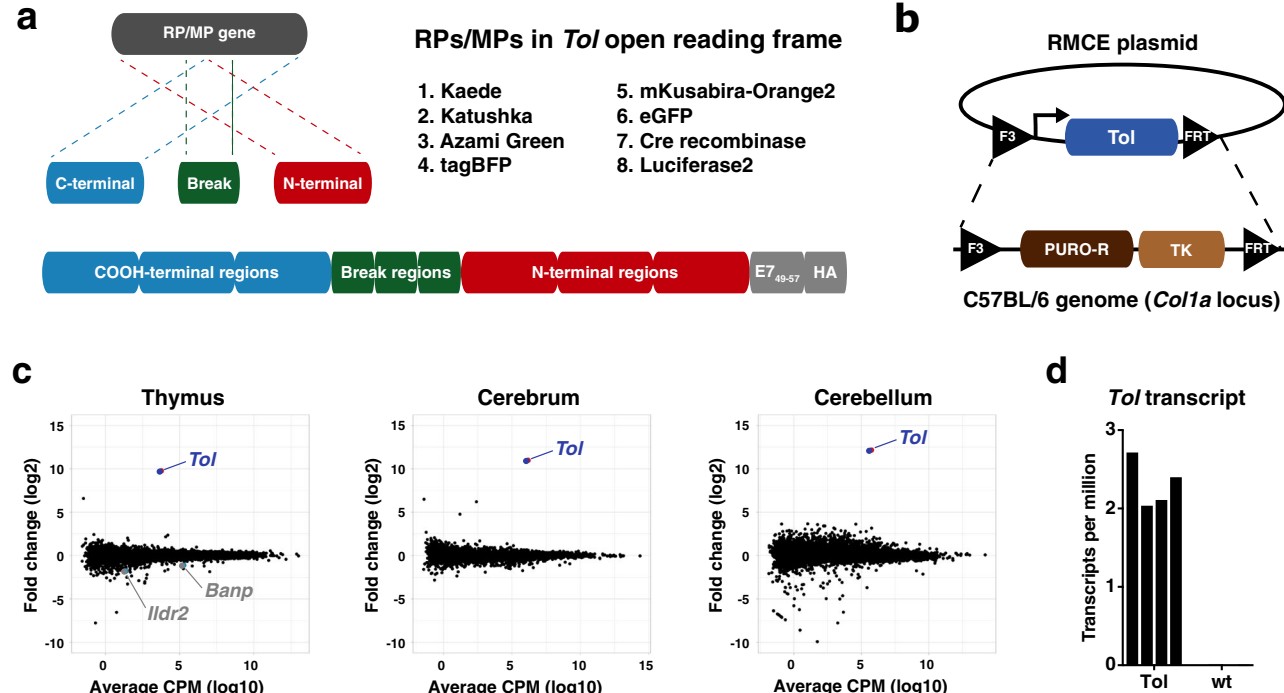

**Fig. 1 Generation and validation of the Tol mouse model. a** Top left: cartoon depicting the strategy for reporter gene fragmentation, applied for each RP/MP gene included in the *Tol* ORF. Top right: reporter proteins included in the *Tol* ORF. Bottom: schematic overview of gene fragment placement in the *Tol* ORF. Note that the *Tol* ORF encodes the HPV $E7_{49-57}$ epitope in the COOH-terminal region of the chimeric Tol protein. **b** Cartoon depicting the targeting strategy of the *Tol* ORF into the *Col1a1* locus through recombinase-mediated cassette exchange. **c** Differential gene expression analysis of indicated organs of Tol mice relative to WT mice. Scatterplots indicate average log2 fold changes and average counts per million. Genes with a statistically significant change in expression in Tol mice are indicated in gray, and the *Tol* transcript is indicated in blue. **d** Expression levels of the *Tol* transcript in thymi of Tol and WT mice. Data shown (**c**, **d**) is aggregated from $n = 4$ (Tol) and $n = 4$ (WT) mice. RP reporter protein, MP modifier protein, RMCE recombinase-mediated cassette exchange, PURO-R puromycin resistance gene, TK thymidine kinase gene, CPM counts per million.

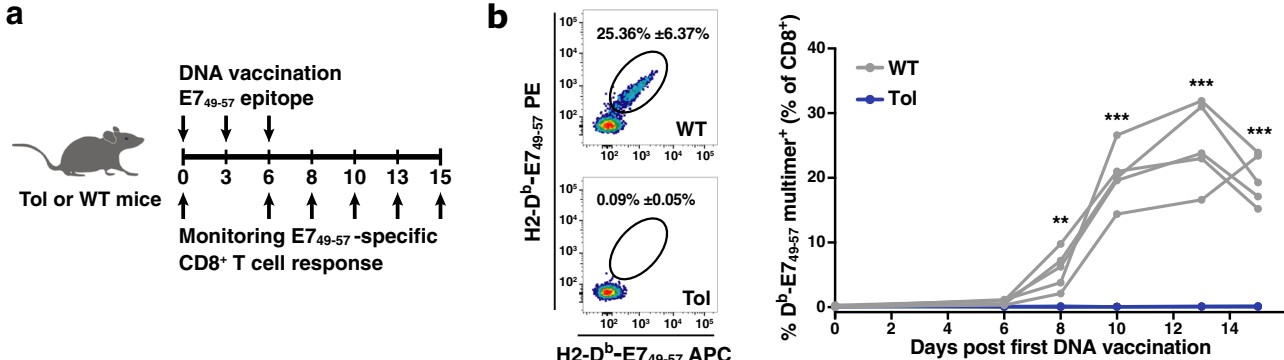

**Fig. 2 Complete loss of $E7_{49-57}$ immunogenicity in Tol mice. a** Schematic overview of vaccination and monitoring strategy. **b** Left: Representative flow cytometry plots depicting $E7_{49-57}$-specific CD8+ T cells at day 13 post DNA vaccination in WT and Tol mice. Right: $E7_{49-57}$-specific CD8+ T-cell response in blood of WT (gray) and Tol (blue) mice at the indicated timepoints post first DNA vaccination. Lines indicate individual mice. Cells are gated on IR-dye− CD8+ lymphocytes. Depicted data are representative of two independent experiments, including at least 5 mice per group. *P* values were determined by Repeated Measures two-way ANOVA with Sidak's multiple comparisons test to test the significance at each timepoint. *$p < 0.05$; **$p < 0.01$; ***$p < 0.001$.

were transduced with a retrovirus encoding the green-to-red photo-switchable reporter protein Kaede. The resulting Kaede+ and Kaede− OT-I T cells were then transferred into Tol and WT mice, and OT-I T-cell frequencies were boosted by administration of an $OVA_{257-364}$-epitope encoding DNA vaccine (Fig. 3a). In the first days following adoptive cell transfer, the numbers of Kaede+ T cells increased rapidly and similarly in both WT and Tol mice (day 10, ratio Tol/WT = 1.25, $p = 0.421$, Fig. 3b). Notably, at later timepoints Kaede+ OT-I T cells formed a readily detectable memory T-cell population in Tol mice, but declined to near undetectable numbers in WT mice (day 57, ratio Tol/WT = 5.6, $p = 0.008$, Fig. 3b,c), suggestive of immunological rejection of the introduced Kaede+ cells. As a control, Kaede− T cells were detected in comparable numbers in WT and Tol mice (day 57, ratio Tol/WT = 1.5, $p = 0.309$). In addition, remaining Kaede+ T cells in WT mice showed a significantly lower expression of the Kaede protein than Kaede+ T cells present in Tol mice ($p = 0.008$, Fig. 3d), indicating that immune-mediated rejection can both lead to a reduction in cell quantities and selection of cells with lower transgene expression. To assess whether expression of the *Tol*

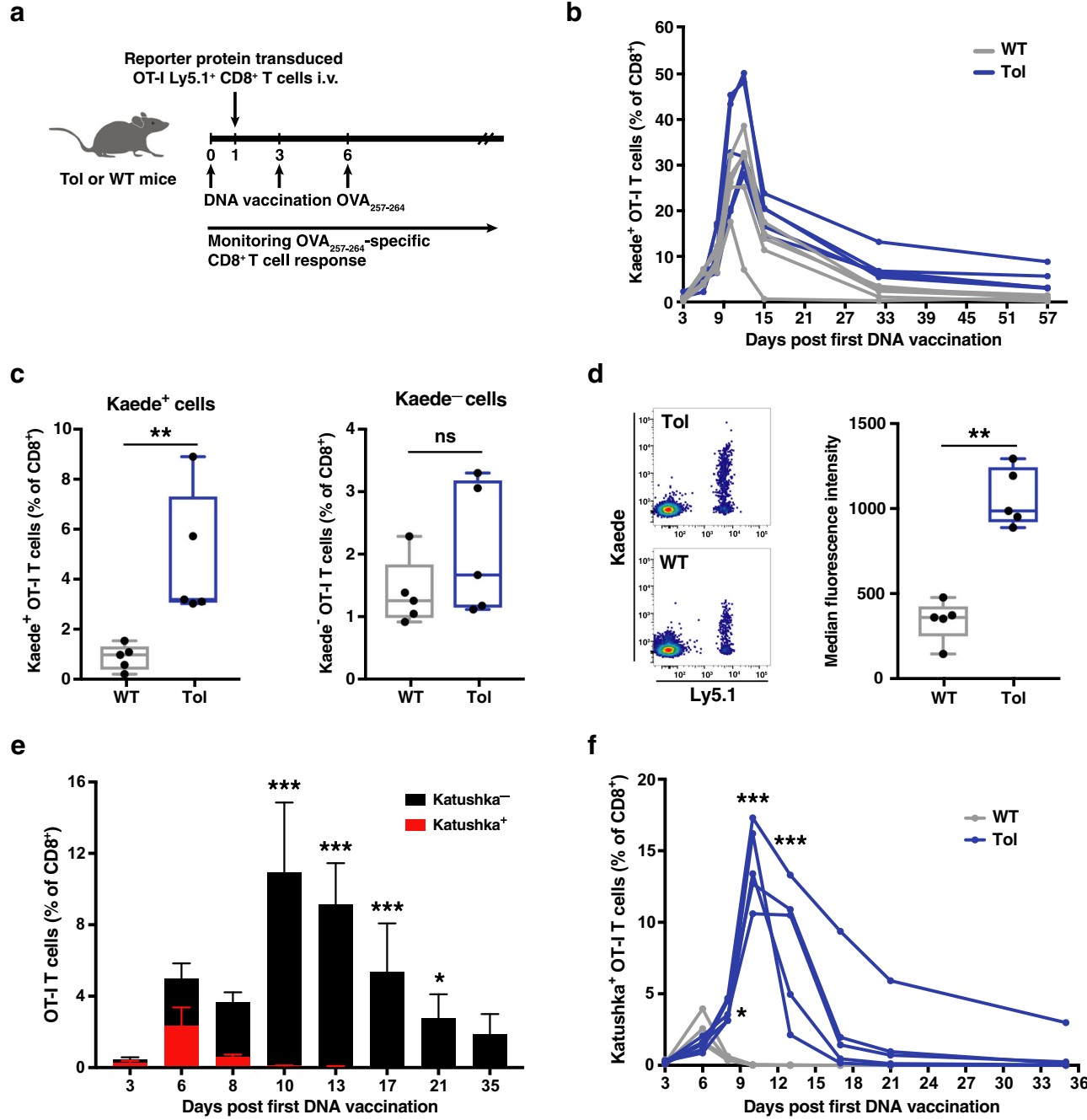

**Fig. 3 Enhanced engraftment of Kaede- and Katushka-expressing cells in Tol mice. a** Schematic overview of experimental design. **b** Percentage of Kaede+ OT-I T cells in the blood of WT (gray) and Tol (blue) mice at the indicated timepoints post first DNA vaccination. **c** Percentage of Kaede+ (left) or Kaede− (right) OT-I cells in WT and Tol mice at day 57 after first DNA vaccination. **d** Left: representative flow cytometry plots depicting Kaede expression in transferred (Ly5.1+) cells at day 57 after first DNA vaccination. Right: median fluorescence intensity (MFI) of Kaede+ OT-I cells in WT and Tol mice at day 57 post first DNA vaccination. **e** Percentage of Katushka+ (red) and Katushka− (black) OT-I cells in WT animals at the indicated timepoints after first DNA vaccination. **f** Percentage Katushka+ OT-I cells in WT and Tol mice at the indicated timepoints after first DNA vaccination. Cells (**b–f**) are gated on IR-dye−CD8+ lymphocytes, and plots depict n = 5 (Tol) and n = 5 (WT) mice. Depicted data are representative of one (**b–d**) or two (**e, f**) independent experiments. Box-plots (**c, d**) depict interquartile range, with whiskers representing minimum and maximum values. Dots (**c, d**) and lines (**b, f**) indicate individual mice. Error bars (**e**) represent SD. P values were determined by either two-tailed Mann–Whitney signed rank test (**c, d**) or Repeated Measures two-way ANOVA with Sidak's multiple comparisons test (**e, f**). *p < 0.05; **p < 0.01; ***p < 0.001, ns not significant.

transgene resulted in long-term tolerance toward Kaede, recipients of Kaede+ and Kaede− OT-I T cells were vaccinated and subsequently challenged by a secondary vaccination 155 days post adoptive cell transfer (Supplementary Fig. 3a). In Tol mice, Kaede+ cells again expanded to large numbers after receiving a secondary stimulus, emphasizing ongoing tolerance toward the

transferred cells (Supplementary Fig. 3b). In contrast, Kaede+ T cells exhibited a stunted proliferative burst, and expressed only low levels of the fluorescent protein in WT animals (Supplementary Fig. 3c). The inability to mount a robust secondary response was directly related to Kaede expression, as Kaede− T cells were able to expand in WT mice (Supplementary Fig. 3d).

We next examined the engraftment potential of OT-I T cells that expressed a third foreign entity, the rapid-folding red-fluorescent protein Katushka, in Tol and WT recipient mice. Adoptive transfer of Katushka⁺ cells induced a vigorous anti-fluorochrome immune response in WT mice, as reflected by the rapid disappearance of Katushka⁺ OT-I T cells, but not Katushka⁻ T cells (ratio Kat⁻/Kat⁺ cells: day 6 = 1.2; day 10 = 385.4, Fig. 3e). Importantly, the observed clearance of Katushka-expressing cells was abrogated by Tol expression, as shown by the presence of an approximately 500-fold larger pool of Katushka⁺ T cells in Tol mice as compared to WT littermates at day 10 post first DNA vaccination ($p = 0.008$, Fig. 3f).

To evaluate whether cells expressing any of the additional FPs encoded by the Tol transgene could engraft in Tol mice, we concurrently transferred OT-I T cells that had been transduced with four different FPs (Katushka, BFP, AzamiGreen or mKO2) into either Tol or WT mice. In line with the prior analyses, Katushka⁺ OT-I T cells disappeared from the circulation of WT mice as early as day 10 post-transfer, whereas robust engraftment of Katushka-expressing cells was observed in Tol mice (Supplementary Fig. 4). Analysis of modified T-cell frequencies for the other three FPs showed the presence of FP-modified cells in blood of both WT and Tol mice, with a trend toward enhanced engraftment in Tol mice (Supplementary Fig. 4). Thus, cells modified with 5 different FPs (Kaede, Katushka, BFP, Azami-Green or mKO2) stably engraft in Tol mice, whereas engraftment of cells expressing at least 2 of these transgenes is severely impaired in WT mice.

## Discussion

Manipulation and detection of cellular pathways using modifier and reporter proteins forms a cornerstone of animal research; however, the xenogenic nature of these proteins introduces a substantial risk of recognition by the adaptive immune system. In this work, we describe an approach to fuse multiple RPs/MPs into a large shuffled ORF, thereby perturbing their functionality while maintaining all potentially immunogenic epitopes. Importantly, we show that this approach allows expression in the thymus plus peripheral tissues, that the encoded fusion protein is fully translated, and that this protein translation confers immunological tolerance toward all epitopes tested. Interestingly, comparison of the fate of Katushka⁺ and Kaede⁺ cells in WT mice underlines that immune recognition can either lead to the rapid clearance of transgene expressing cells (i.e., Katushka), or to the gradual selection of cells with lower transgene expression (i.e., Kaede). This second mode of immune rejection may go unnoticed in many experimental settings, thereby representing a hidden confounder. Engraftment of BFP, AzamiGreen and mKO2 transduced OT-I T cells was also observed but was only marginally improved relative to WT mice, suggesting that these proteins have a low immunogenicity in the C57BL/6 strain used in this study. As antigen presentation is MHC restricted, the value of the Tol transgene for these FPs may be more profound in other mouse strains, as e.g., exemplified by the preferential immunogenicity of eGFP in Balb/c mice relative to C57BL/6 strains[5,16,17].

Our study provides a methodological framework through which shuffled transgenes are applied to avoid undesirable immune responses against exogenously introduced proteins. Both the Tol ORF and the C57BL/6 Tol strain used in this work are available upon reasonable request (from the corresponding author and the Netherlands Cancer Institute Transgenics Core Facility, respectively). Furthermore, creation of a resource of mouse strains that are tolerant for additional reporter and modifier proteins used in biomedical research would be of considerable value.

## Methods

**Generation of the Tol open reading frame and vaccination plasmids.** The Tol ORF was designed as described in Supplementary Fig. 1 and synthesized by Genscript. The full-length ORF was subcloned into a recombinase-mediated cassette exchange (RMCE) compatible vector, producing the pF3-CAG-Tol-FRT plasmid (Addgene ID: 141349). In the resultant vector, the Tol ORF was positioned downstream of the CAG promoter and flanked by F3 and FRT recombination sites. The entire F3-CAG-Tol-FRT cassette was sequence verified by Sanger sequencing. Katushka, Kaede, AzamiGreen, mKO2 and tagBFP ORFs were shuttled directly into the multiple cloning site of pMP71 via Gibson cloning to generate pMP71-Katushka (Addgene ID: 141351), pMP71-Kaede (Addgene ID: 141352), pMP71-AzamiGreen (Addgene ID: 141353), pMP71-mKO2 (Addgene ID: 141354) and pMP71-tagBFP (Addgene ID: 141355), respectively. Generation of pVAX- E7₄₉₋₆₇ has been described previously[18]. pVAX-SIINFEKL (Addgene ID: 141350) was generated by Gibson cloning to produce scarless fusions between SIINFEKL, several CD4⁺ T-cell epitopes (HELP), and the KDEL-signal peptide.

**Mice.** C57BL/6-Ly5.1 and OT-I mice were obtained from Jackson Laboratories and crossed to obtain C57BL/6-Ly5.1-OT-I donor mice for adoptive transfer experiments. All animals were maintained and bred in the animal department of The Netherlands Cancer Institute and used for experimentation at 7–14 weeks. All animal experiments were approved by the Animal Welfare Committee of the NKI, in accordance with national guidelines.

**Generation of Tol transgenic mice.** The Tol expression cassette was introduced into a locus 3′ to the Col1a1 locus[14] using recombinase-mediated cassette exchange (RMCE)[19] In brief, a C57BL/6J ES cell line was derived from blastocysts[20] and an F3-Puro-deltaTK-FRT cassette was targeted into the ESCs by homologous recombination, as described[14]. The pF3-CAG-Tol-FRT plasmid was co-transfected with pCAGGS-FLPe into the B6J-RMCE ES cells, followed by selection in medium containing fialuridine. Clones were screened by PCR for correct and complete integration of the F3-Tol-FRT cassette into the Col1a1 locus. Correctly modified ESCs were then injected into B6/NTAC blastocysts, and injected blastocysts were transferred into pseudo-pregnant B6CBAF1/JRj foster mice, as described previously[13,20]. Resulting chimeras were tested for chimerism by quantitative-PCR (Q-PCR). Positive animals (>0.4 RQ) were crossed with C57BL/6JRj mice (Janvier) to generate experimental cohorts. Presence of the Tol cassette was confirmed in experimental cohorts by PCR using the forward 5′-GGAAAGAATCACAACTTACG-3′ and reverse 5′-AGAGCATTTCGG TTGAGGCC-3′ primers. The Tol strain described in this communication is available upon request from the Netherlands Cancer Institute Transgenics Core Facility.

**Histopathology.** For histopathological analyses, 2 μm-thick hematoxylin-eosin stained sections were prepared from formalin-fixed, paraffin-embedded murine tissues such as skin, spleen, thymus, lymph nodes, liver, pancreas, gastrointestinal tract, heart, lung, kidneys, testes, ovaries, accessory sex glands, bone marrows (sternum and extremity), and muscles. Sections were evaluated and scored by an animal pathologist blinded to animal genotype.

**RNA sequencing.** RNA was extracted from the indicated frozen tissues using the RNeasy Mini Kit (Qiagen). Whole transcriptome sequencing samples were prepared with the TruSeq Stranded mRNA Kit (Illumina). Single-end 65 bp sequencing was performed on a HiSeq 2500 System (Illumina). Transcript abundance was calculated using Salmon 0.14.1[21], based on the GRCm38 transcriptome build. The Tol transcript sequence was added before read alignment. Differential gene expression analysis was performed using EdgeR 3.9[22].

**Production of retroviral supernatants.** Phoenix-E packaging cells were seeded at 1.4*10⁶ cells per 10 cm dish in IMDM supplemented with 8% FCS, glutamax, penicillin/streptomycin (Gibco). After 24 h, medium was refreshed and cells were transfected using FuGene6 (Roche), following the manufacturer's protocol. 48 h post transfection, viral supernatant was harvested and passed through a 0.22 μm filter (Sigma). Filtered retroviral supernatant was snap-frozen in liquid nitrogen and stored at −80 °C.

**Generation and adoptive transfer of Kaede, Katushka, BFP, AzamiGreen or mKO2 positive OT-I splenocytes.** Spleens from C57BL/6;Ly5.1;OT-I mice were passed through 70 μm strainers to obtain a single cell suspension. Splenocytes were then seeded at 6*10⁶ cells/ml in RPMI 1640 supplemented with 8% FCS, glutamax, penicillin/streptomycin, pyruvate, non-essential amino acids, HEPES, β-mercapto-ethanol (Gibco), 2 μg/ml concanavalin A (Calbiochem) and 1 ng/ml IL-7 (Pepro-tech). After 48 h, T cells were harvested and re-seeded on 24-well plates coated with Retronectin (Takara) at $1.5 \times 10^6$ cells per well in RPMI 1640 supplemented with 8% FCS, glutamax, penicillin/streptomycin, pyruvate, non-essential amino acids, HEPES, β-mercapto-ethanol and 50 IU IL-2 (Novartis). Retrovirus containing supernatant was added and cells were transduced by spin-infection at 2000 rpm for 90 minutes. After 24 h, cells were harvested and washed twice in PBS. Subsequently, T cells were resuspended in HBSS (Gibco) and $1 \times 10^6$ CD8⁺ cells were intravenously injected via the tail vein.

**DNA vaccination**. One day prior to vaccination with DNA encoding MHC-II class restricted helper epitopes[18], plus either the OVA$_{257-264}$ epitope ('Help-OVA') or the E7$_{49-67}$ epitope ('Help-E7'), hair of mice was removed from hind legs using Veet depilation cream (Reckitt Benckiser). Primary DNA vaccination was subsequently performed on day 0, 3 and 6, as previously described[23]. In brief, a droplet of 15 µl of a 2 µg/µl DNA solution in 10 mmol/L Tris pH 8.0 and 1 mmol/L EDTA pH 8.0 was applied on both the inside and outside of the leg, and vaccination was performed using a rotary tattoo device with a sterile disposable 9-needle bar (MT.DERM) oscillating at a frequency of 100 Hz for 1 min with a needle depth of 1 mm. For secondary vaccinations, mice received a single DNA tattoo with 20 µl of the 2 µg/µl plasmid solution on both the inside and outside of the leg, at >155 days after start of primary vaccination.

**Analysis of antigen-specific T-cell responses by flow cytometry**. Blood samples were collected from the tail vein at the indicated timepoints. Erythrocytes were removed by treatment with NH$_4$Cl buffer, and cells were washed and then stained with anti-CD45.1-APC (A20, Thermo Fisher Scientific), anti-CD8-PerCp-Cy5.5 (eBioH35-17.2, Thermo Fisher Scientific), near-IR dye (Thermo Fisher Scientific) and indicated MHC multimers. MHC multimers were produced in-house by UV-induced ligand exchange and subsequent labeling with BV421, APC or PE (Thermo Fisher Scientific), as described previously[24]. Flow cytometry data were acquired on a Fortessa (BD Biosciences) and analyzed in FlowJo (version 10.4.2) according to the gating strategy shown in Supplementary Fig. 5.

**Statistics and reproducibility**. Statistical analyses were performed in Prism (GraphPad), comparing groups of mice with the two-tailed Mann–Whitney test. Results were regarded as statistically significant at a $P$ value of <0.05. Images were combined and annotated in Illustrator (version CC 2014, Adobe) for presentation.

**Reporting summary**. Further information on research design is available in the Nature Research Reporting Summary linked to this article.

## Data availability

Mouse cDNA sequences (GRCm38) used for Salmon alignment were downloaded from Ensembl (http://www.ensembl.org/biomart/martview). RNA-seq data have been deposited, and are available from the Gene Expression Omnibus under accession number GSE147757. All source data underlying the graphs and charts presented in the main figures are presented in Supplementary Data 3.

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

## Acknowledgements

We would like to thank M. Toebes for generation of MHC multimers, and L. Kroese, R. Bin Ali and T. Braumuller for technical assistance in generation and validation of the Tol mouse model. In addition, we thank the NKI animal facility, animal pathology, flow cytometry and genomics core facility for technical support and members of the Schumacher and Haanen laboratories for discussions. Finally, we would like to thank M. E. Hoekstra for providing illustrations used in the figures. This work was supported by Institute for Chemical Immunology (ICI) grant 003 and ERC AdG Life-His-T (to T.N.S.).

## Author contributions

K.B. generated the Tol targeting vector and other vectors, and performed monitoring of T-cell responses. F.E.D. performed adoptive cell transfer and vaccination experiments, and performed monitoring of T-cell responses. J.-Y.S. evaluated histopathological data. J.C.R., K.B., F.E.D. and F.A.S. contributed to design of the Tol ORF. I.J.H. and C.E.J.P. generated the genetically engineered Tol mouse model. K.B., F.E.D., F.A.S. and T.N.S. contributed to experimental design and prepared the manuscript with input of all co-authors.

## Competing interests

The authors declare no competing interests.
