## [Peer Review File · Communications Biology]

Reviewers' comments:

Reviewer #1 (Remarks to the Author):

Bresser and colleagues describe the generation of a novel "Tol" transgenic mouse that expresses a number of different reporter and modifier proteins in a scrambled format. The authors show that this Tol mouse is (1) unable to respond to antigens expressed on the transgene and (2) able to allow survival of adoptively transferred fluorescent cells that are routinely rejected in wild-type mice. The model is validated appropriately and will be useful for studies using adoptive transfer of cells expressing the relevant reporter and/or modifier proteins. It is likely that those researchers using such an approach will be interested in this study and may employ this tool in their research.

Reviewer #2 (Remarks to the Author):

Rejection of cells and tissues expressing reporters or suppression if their expression with selection of cells with low levels of reporter expression is a major problem in biomedical research. Authors suggest an original approach to resolve this problem, at least for some of the reporters, here a panel including Kaede, Katushka, Azami Green, tagBFP, mKusabira-Orange2, eGFP, Cre recombinase and Luciferase 2. Authors created a "Tol" transcript encoding reshuffled reporters, each represented by N-terminal, C-terminal and break region, placed all C-terminal first, then all breaks, then all N-terminals. Thus, none of the reporters were functional, and all were very well represented to the immune system of the mice. Tol gene was targeted to Col1a1 locus of embryonic stem cells via recombinase mediated cassette exchange. Modified embryonic stem cells were injected into blastocysts and transferred to pseudopregnant foster mice. Mice were proven to express Tol in different organs and tissues, however, the major expression profiles of Tol mice compared to WT mice did not change, and expression of Tol despite its unfolded status and possible induction of unfolded protei response did not induce any pathologies. Furthermore, Tol ORF included a reporter epitope on the C-terminus derived from HPV 16 E7, unabling to assess if Tol was truly tolerated. Indeed, Tol mice built no reponse against reporter epitope as compared to WT mice DNA-immunized with HPV 16 E7 epitope encoding construct. Tolerance to reporters was then demonstrated by the engraftment of Kaede and Katushka expressing cells into Tol and WT mice. Challenge cells were CD8+ T cells specific for OVA peptide, their quantity was accurately assessed by flow cytometry. After the challenge, both Tol and WT mice were DNA immunized with OVA peptide to increase frequency of transplanted OVA specific T-cells. The frequency significantly increased in Tol, but not in WT mice that showed only stunned boost, proving the induction of tolerance to Kaede and Katushka. Authors noted an interesting phenomenon of differential clearance or suppression of reporter expressing cells in WT mice – either by killing of reporter expressing cells (Katushka), or by lowering the levels of reporter expression (Kaede). Ideology is very interesting and productive, tolerization proiving experiments are elegantly done, article is well written. Could be accepted after minor revision.

Major comments

1. This article is a methodology that can be applied to any reporter, its functionality for two out of six, Kaede and Katushka, was confirmed. It was not shown what would be the immune response against the other reporter proteins, like eGFP known to be immunogenic. On the other hand, luciferases have low immunogenicity, their inclusion into Tol ORF was not motivated. Now it is included, but we do not know what would be the response to Luc2 in WT mice, how much it is reduced in Tol mice, and what is the mechanism for tolerance – clearance of expressing cells or downregulation of expression. A simple readout test for induction of immune response against other four reporters needs to be included, such as CD8+ T cell response against a known dominant CTL epitope recognized in each of the four uncharacterised reporter proteins, recognized in C57BL/6- or OT-1 mice, if authors have frozen cells left. Otherwise, authors need to point out that responses against other reporters were not assessed.

2. Rejection of reporter-expressing cells as presented by the authors solely depends on CD8+ T cell response. CD4+ T cells are known to have lytic activity as well. CD4+ T cell driven rejection needs to be considered, at least as a possibility and a pathway of evade.

Minor comments

1. Scheme representing Tol in Figure 1 is confusing. It is schematic, made for three reporter moieties (any), listing actual ones in the panel legend. Additional figure with a complete list of moieties is given in Supplement Fig 1. One scheme of the Suppl Fig 1 would be sufficient.

2. Fig 1b – schematic representation of vaccination and testing of T cell response is unclear. X-axis should show days, and indicate what is done when – immunization at 0, 3, 6 days and tests on day 15 (with arrows or likewise).

3. Fig 1d shows response (% responsive CD8+ T cells) to HPV 16 E7 epitope in WT and Tol mice – 2 and 3 as representative of 5 in each group. There are only 10 mice all in all, why not show all, possibly split into two panels, or as supplement? (as is done for Katushka+ OT-I T cells in Fig 2f). Also significance of difference between WT and Tol mice in CTL response to E7 epitope needs to be shown for the given days (6, 8, 13, 15).

4. DNA immunization protocol contains unclarities. Mice are primed by DROPLET (15 ul of 2 ug/ml DNA solution) – on inside and outside of the leg. Is it one drop split into two portions, or one drop on inside and one on outside (ie 2 times higher DNA dose)? In secondary immunization, mice received a single DNA tattoo of 20 ul of 2 ug/ul DNA – indicating that priming application indeed contained two doses, but this needs to be specified. For tattooing device, one needs to provide the brand and the manufacturer.

Response to referees

General remarks:

We are grateful to the reviewers for their positive and constructive feedback on our manuscript “A multiple reporter protein tolerant mouse model”. We were happy to see that both reviewers consider our work of substantial interest and well performed. We appreciate their constructive criticism, and feel that the incorporation of their suggestions has significantly improved the current manuscript. Below, we have addressed the comments of both reviewers.

Reviewer comments:

Reviewer #1 (Remarks to the Author):

Bresser and colleagues describe the generation of a novel “Tol” transgenic mouse that expresses a number of different reporter and modifier proteins in a scrambled format. The authors show that this Tol mouse is (1) unable to respond to antigens expressed on the transgene and (2) able to allow survival of adoptively transferred fluorescent cells that are routinely rejected in wild-type mice. The model is validated appropriately and will be useful for studies using adoptive transfer of cells expressing the relevant reporter and/or modifier proteins. It is likely that those researchers using such an approach will be interested in this study and may employ this tool in their research.

Reply: We thank reviewer #1 for careful reading of our work, and are happy to see the reviewer feels that our work is properly executed and will provide a useful tool in the field.

Reviewer #2 (Remarks to the Author):

Rejection of cells and tissues expressing reporters or suppression if their expression with selection of cells with low levels of reporter expression is a major problem in biomedical research. Authors suggest an original approach to resolve this problem, at least for some of the reporters, here a panel including Kaede, Katushka, Azami Green, tagBFP, mKusabira-Orange2, eGFP, Cre recombinase and Luciferase 2. Authors created a “Tol” transcript encoding reshuffled reporters, each represented by N-terminal, C-terminal and break region, placed all C-terminal first, then all breaks, then all N-terminals. Thus, none of the reporters were functional, and all were very well represented to the immune system of the mice. Tol gene was targeted to Coll1a1 locus of embryonic stem cells via recombinase mediated cassette exchange. Modified embryonic stem cells were injected into blastocysts and transferred to pseudopregnant foster mice. Mice were proven to express Tol in different organs and tissues, however, the major expression profiles of Tol mice compared to WT mice did not change, and expression of Tol despite its unfolded status and possible induction of unfolded protein response did not induce any pathologies. Furthermore, Tol ORF included a reporter epitope on the C-terminus derived from HPV 16 E7, unabling to assess if Tol was truly tolerated. Indeed, Tol mice built no reponse against reporter epitope as compared to WT mice DNA-immunized with HPV 16 E7 epitope encoding construct. Tolerance to reporters was then demonstrated by the engraftment of Kaede and Katushka expressing cells into Tol and WT mice. Challenge cells were CD8+ T cells specific for OVA peptide, their quantity was accurately assessed by flow cytometry. After the challenge, both Tol and WT mice were DNA immunized with OVA peptide to increase frequency of transplanted OVA specific T-cells. The frequency significantly increased in Tol, but not in WT mice that showed only stunned boost, proving the induction of tolerance to Kaede and Katushka. Authors noted an interesting phenomenon of differential clearance or suppression of reporter expressing

cells in WT mice – either by killing of reporter expressing cells (Katushka), or by lowering the levels of reporter expression (Kaede).

Ideology is very interesting and productive, tolerization proving experiments are elegantly done, article is well written. Could be accepted after minor revision.

Major comments 1. This article is a methodology that can be applied to any reporter, its functionality for two out of six, Kaede and Katushka, was confirmed. It was not shown what would be the immune response against the other reporter proteins, like eGFP known to be immunogenic. On the other hand, luciferases have low immunogenicity, their inclusion into Tol ORF was not motivated. Now it is included, but we do not know what would be the response to Luc2 in WT mice, how much it is reduced in Tol mice, and what is the mechanism for tolerance – clearance of expressing cells or downregulation of expression. A simple readout test for induction of immune response against other four reporters needs to be included, such as CD8+ T cell response against a known dominant CTL epitope recognized in each of the four uncharacterized reporter proteins, recognized in C57BL/6- or OT-1 mice, if authors have frozen cells left. Otherwise, authors need to point out that responses against other reporters were not assessed.

Reply: With respect to this first comment of reviewer #2, we had indeed restricted our experimental validation in the C57bBL/6 strain to two fluorescent proteins plus the carboxy terminal HPV-derived epitope. As we observe tolerance in 3 out of 3 cases, we respectfully felt that this was likely to extend to other epitopes. With respect to T cell recognition of luciferase, the jury is still out regarding the immunogenicity of this protein in the C57BL/6 strain, with clear evidence for T cell reactivity observed in some experimental settings (1). Of more importance, even if luciferase (or any of the other transgene products) would be fully non-immunogenic in C57BL/6 mice, this would not make inclusion superfluous. Specifically, we developed the *Tol* transgene cassette as a technology to allow engraftment of fluorescently modified cells in different mouse strains with diverse MHC haplotypes. To highlight the relevance of this, immune reactivity against luciferase has in fact been shown to limit engraftment of Luc expressing cells in Balb/c mice (2).

In the revised manuscript we clarify that, while not all transgenes will necessarily be immunogenic in C57BL/6 mice, their inclusion is motivated by the fact that different foreign proteins are known to be immunogenic in different mouse strains. In addition, we have clarified that we are making the *Tol* cassette available for other researchers, for incorporation in their preferred mouse models (page 5 of the revised text).

Finally, recognizing that it would indeed be useful to know whether the other fluorescent proteins encoded by the *Tol* transgene would induce immune rejection in WT C57BL/6 mice, we directly compared the fate of infused cells that either express Katushka, BFP, AzamiGreen or mKO2 in WT and Tol mice. The resulting data, demonstrating that Katushka is the protein with the highest immunogenicity in C57BL/6 mice of this set, have also been included in the revised manuscript.

Major comments 2. Rejection of reporter-expressing cells as presented by the authors solely depends on CD8+ T cell response. CD4+ T cells are known to have lytic activity as well. CD4+ T cell driven rejection needs to be considered, at least as a possibility and a pathway of evade.

Reply: We very much agree with the reviewer that CD4+ T cell responses can contribute to rejection of cells. The observation that the *Tol* transgene prevents rejection of cells that either

express the Kaede or the Katushka fluorescent protein is consistent with the idea that the *Tol* transgene induces tolerance in both the CD8⁺ and the CD4⁺ T cell lineage. However, we very much agree that we do not know whether this would involve thymic deletion or peripheral tolerance. Following the suggestion of the reviewer, we have modified the introduction of the manuscript to indicate the potential role of CD4⁺ T cells, also referring to the relevant literature on this topic, which shows that for systemically expressed self-antigens (as is the case for *Tol*), tolerance induction generally occurs through deletion of high affinity CD4⁺ T cells, with a low affinity CD4⁺ T cell population that is also less responsive to antigen remaining (3,4).

Minor comments 1. Scheme representing Tol in Figure 1 is confusing. It is schematic, made for three reporter moieties (any), listing actual ones in the panel legend. Additional figure with a complete list of moieties is given in Supplement Fig 1. One scheme of the Suppl Fig 1 would be sufficient.

Reply: We agree with reviewer #2 that Figure 1a and Supplementary figure 1 provide partial redundant information. We intended Fig 1a to serve as a cartoon depiction of the shuffling strategy resulting in the *Tol* ORF, rather than an exact depiction of each segments' placement (which is shown in Suppl. Fig. 1). In response to the reviewer's comment we have edited Fig 1a to improve intelligibility. In addition, we have made changes to the figure legend that should increase clarity.

Minor comments 2. Fig 1b – schematic representation of vaccination and testing of T cell response is unclear. X-axis should show days, and indicate what is done when – immunization at 0, 3, 6 days and tests on day 15 (with arrows or likewise).

Reply: We have revised all experimental setup diagrams (Fig 2a, Fig 3a, Supplementary Fig 3a) following the suggestions of the reviewer.

Minor comments 3. Fig 1d shows response (% responsive CD8+ T cells) to HPV 16 E7 epitope in WT and Tol mice – 2 and 3 as representative of 5 in each group. There are only 10 mice all in all, why not show all, possibly split into two panels, or as supplement? (as is done for Katushka+ OT-I T cells in Fig 2f). Also significance of difference between WT and Tol mice in CTL response to E7 epitope needs to be shown for the given days (6, 8, 13, 15).

Reply: Fig 1d (Fig 2b in revised manuscript) does depict all 10 mice (1 line per mouse). All 5 WT mice (grey) are shown to respond to DNA tattoo. In contrast, all 5 Tol mice (blue) do not respond to DNA tattoo at all, making it difficult to distinguish these curves from the x-axis. In the revised manuscript, we have amended the graph to make the different lines more visible (increased contrast between grey and blue, and lowered the x-axis).

In addition, following the referee's suggestion, we have incorporated Repeated Measures ANOVA's for each time course experiment (Fig 2b, Fig 3b, e, f) in the manuscript, and added significance values into the figures.

Minor comments 4. DNA immunization protocol contains unclarities. Mice are primed by DROPLET (15 ul of 2 ug/ml DNA solution) – on inside and outside of the leg. Is it one drop split into two portions, or one drop on inside and one on outside (ie 2 times higher DNA dose)? In secondary immunization, mice received a single DNA tattoo of 20 ul of 2 ug/ul

DNA – indicating that priming application indeed contained two doses, but this needs to be specified. For tattooing device, one needs to provide the brand and the manufacturer.

Reply: The methods section describing the immunization protocol has been clarified, as suggested.

References

1: Limberis MP, Bell CL, Wilson JM. Identification of the murine firefly luciferase-specific CD8 T-cell epitopes. *Gene Ther.* 2009 Mar;16(3):441-7. doi: 10.1038/gt.2008.177. Epub 2009 Jan 8. PubMed PMID: 19129859.

2: Baklaushev VP, Kilpeläinen A, Petkov S, Abakumov MA, Grinenko NF, Yusubalieva GM, Latanova AA, Gubskiy IL, Zabozaev FG, Starodubova ES, Abakumova TO, Isaguliants MG, Chekhonin VP. Luciferase Expression Allows Bioluminescence Imaging But Imposes Limitations on the Orthotopic Mouse (4T1) Model of Breast Cancer. *Sci Rep.* 2017 Aug 10;7(1):7715. doi: 10.1038/s41598-017-07851-z. PubMed PMID: 28798322; PubMed Central PMCID: PMC5552689.

3: Moon JJ, Dash P, Oguin TH 3rd, McClaren JL, Chu HH, Thomas PG, Jenkins MK. Quantitative impact of thymic selection on Foxp3+ and Foxp3- subsets of self-peptide/MHC class II-specific CD4+ T cells. *Proc Natl Acad Sci U S A.* 2011 Aug 30;108(35):14602-7. doi: 10.1073/pnas.1109806108. Epub 2011 Aug 22. PubMed PMID: 21873213; PubMed Central PMCID: PMC3167500.

4: Legoux FP, Lim JB, Cauley AW, Dikiy S, Ertelt J, Mariani TJ, Sparwasser T, Way SS, Moon JJ. CD4+ T Cell Tolerance to Tissue-Restricted Self Antigens Is Mediated by Antigen-Specific Regulatory T Cells Rather Than Deletion. *Immunity.* 2015 Nov 17;43(5):896-908. doi: 10.1016/j.immuni.2015.10.011. Epub 2015 Nov 10. PubMed PMID: 26572061; PubMed Central PMCID: PMC4654997.

REVIEWERS' COMMENTS:

Reviewer #2 (Remarks to the Author):

Manuscript has been revised in lines with the suggestions, and can be accepted for publication in the present form.